# Do Health Insurance Schemes Heterogeneously Affect Income and Income Distribution? Evidence from Chinese Agricultural Migrants Survey

**DOI:** 10.3390/ijerph17093079

**Published:** 2020-04-28

**Authors:** Xiaojun Lu, Qun Wang, Daishuang Wei

**Affiliations:** Faculty of Humanities and Social Sciences, Dalian University of Technology, Dalian 116024, China; lxj79@dlut.edu.cn (X.L.); daishuang.wei@mail.dlut.edu.cn (D.W.)

**Keywords:** health insurance, income, agricultural migrants

## Abstract

Currently, the particularity of Chinese agricultural migrants groups determines that they can participate in various types of public health insurance schemes, i.e., the New Cooperative Medical Scheme (NCMS), Urban Residents Basic Medical Insurance (URBMI), and Urban Employees Basic Medical Insurance (UEBMI). The goal of this paper is to shed light on whether and how these health insurance schemes affect the agricultural migrants’ income and income distribution. A dataset of 86,660 individuals is obtained from China Migrants Dynamic Survey implemented by the National Health Commission. The study uses the basic ordinary least squares regression to assess association between health insurance schemes and income and uses the propensity score matching method to estimate the income effect. In addition, we further use the quantile regression method to explore heterogeneous effects of health insurance schemes on income distribution. We find that UEBMI and URBMI have significant increased monthly net income of agricultural migrants, while NCMS does not. The income-increasing effect of UEBMI is greater than that of URBMI. The income-increasing effect of UEBMI is most obvious in the low-income group. While URBMI has a significant role in increasing income with its income-increasing effect being obvious for the lowest and highest income groups. We suggest that China’s health insurance system needs further reforms in order to reduce income inequality of agricultural migrants.

## 1. Introduction

One of the fundamental stated goals of public health insurance is to reduce the impact of health shocks on individual income and to reduce poverty due to illness for overall population, especially the poor subgroups [1]. After more than 20 years of efforts, China built up a universal health insurance system in 2015. This system contains three public health insurance schemes: the Urban Employees Basic Medical Insurance (UEBMI, covering the urban employees and financed by employers and employees), the New Cooperative Medical Scheme (NCMS, covering the rural population and financed by government subsidy and individual premium), and the Urban Residents Basic Medical Insurance (URBMI, covering the urban unemployed and also financed by government subsidy and individual premium). However, China’s public health insurance system still face quite some challenges with one of them being inequity among the three health insurance schemes [2]. To further reform the system, the information on whether and how these different health insurance schemes affect citizens’ income and income distribution is of importance to policy makers.

During these two decades, China is also undergoing a rapid urbanization process. More and more agricultural migrants have entered cities to find employment opportunities. Till the end of 2019, China has more than 290 million agricultural population, which approximately accounted for 20% of the total population [3]. Though agricultural migrants work in the city, they do not have a permanent citizenship. Such identity particularity allows them to participate in NCMS, URBMI, or UEBMI, which provides an empirical basis for us to explore the effects of different health insurance schemes on income and income distribution.

### Health Insurance, Income, and Equity

Health insurance influences household income through two channels: Improvement of health status and reduction in uncertainty of healthcare expenditure. On one hand, health insurance can lower the price of medical service and thus may stimulate the outpatient and inpatient utilization. Therefore, health insurance may lead to the improvement of health status [1,4]. The recovery and improvement of health has “augmenting” and “stabilization” effects on income [5]. Augmenting effects refer to higher labor efficiency and more actual labor supply thanks to the improvement of health. While stabilization effects refer to less income loss since the improvement of health may reduce health service utilization and thus decease direct and indirect illness costs. On the other hand, health insurance may reduce uncertainty of healthcare expenditure and thus may increase household investment in high return [6,7].

Most studies on the impact of health insurance focused on health care utilization, out-of-pocket expenditures, and health status [1,8,9,10,11,12]. Another large group of studies on the impact evaluation of health insurance were centered on health payment-induced poverty reduction and benefit distribution. NCMS was found to have limited financial protection and did not play a significant role in reducing health payment-induced poverty [13,14]. Some researchers found that the rich reaped more from NCMS and URBMI than the poor in China [15,16], while others found that NCMS became more pro-poor after the new round of reforms [4]. As mentioned above, health insurance influences income by multiple mechanisms. However, very limited studies have focused on such topic and the very few studies were all on the single health insurance effect and had mixed conclusions. Hamid et al. revealed that the participation of micro health insurance did not significantly influence household per capita income [5]. Huang found that participation in URBMI significantly increased the household per capita income growth by 13.78% [17]. Qi demonstrated that NCMS significantly raised per capita incomes by 4% on average, but did not influence income distribution within a province [18]. These studies focused on the average income effect of health insurance, but did not further analyze the income distribution effect of health insurance for subgroups. Meanwhile, to our knowledge, very few researchers have paid attention on the heterogeneous effect of different health insurance schemes on the related outcomes, especially income. As mentioned above, agricultural migrants in China have heterogeneous accesses to the three public health insurance schemes and are characterized with semi-citizenization, non-territoriality, and high-mobility. The health insurance effect on income among this special group, agricultural migrants in China, must be very interesting and policy relevant. However, such study is very scarce in China. This paper is going to filling this gap by using a nationwide survey among Chinese agricultural migrants. We will explore the heterogeneous effects of health insurance schemes on income and income distribution among agricultural migrants in China. 

## 2. Materials and Methods 

### 2.1. Dataset

We used the data from the 2017 wave of China Migrants Dynamic Survey implemented by the National Health Commission. The link of the data is as follows: http://www.chinaldrk.org.cn/wjw/#/data/classify/population/yearList. The data collection was organized by the Department of Migrant Population of the National Health Commission of China. This nationwide survey involved around 140,000 migrants from 31 provinces in China with detailed information on individual demographics, income, health insurance, health service utilization, and so on. Agricultural migrants in this study were defined as the people whose household registration was in the rural areas, but living in the urban areas for more than one month. In order to measure the income effect of a single type of public health insurance scheme effectively, we ruled out those who had more than one public health insurances and included 86,660 individuals in the final sample. All the survey sites were urban areas.

### 2.2. Explained and Explanatory Variables

Table 1 briefly summarizes all relevant variables, with means and standard deviations. The dependent variable was defined as an agricultural migrant’s monthly net income in this paper. It was expressed in natural logarithmic transformation. The whole sample was divided into six groups by monthly net income.

The key independent variables were expressed in three dummy variables to indicate whether agricultural migrants participated in NCMS, URBMI, or UEBMI. The other explanatory variables, i.e., control variables, were classified into three categories: Individual characteristics, migration characteristics, and health characteristics. Based on previous studies, age, gender, education, marital status, and employment status were included as individual characteristics factors in this study [19,20,21]. The migration range variable was referred to as whether the individual migrated within the province or out of the province where he/she originally resided. And the health characteristics variable was defined as self-reported health status. In order to eliminate the influence of outliers, Winsorize tailing was applied for continuous variables in this study, i.e., age and monthly net income, and the ratio was set to 1% [22]. 

### 2.3. Empirical Analysis 

First, we used a basic ordinary least squares (OLS) multiple regression model to assess whether health insurance can improve the individual’s income level. The generic equation is expressed as follows:(1)$$LnY_{i}=\alpha +\beta insur_{i}+\gamma ic_{i}+\delta fc_{i}+\eta hc_{i}+\varepsilon$$
where ***Y****_i_* is agricultural migrant *i*’s monthly net income (*i* = 1, 2, ……, 86,660), and we transformed monthly net income into natural logarithm and made it close to a normal distribution [23]; ***insur**_i_* is agricultural migrant *i*’s heterogeneous health insurance participation situation; ***ic**_i_* is the vector of agricultural migrant *i*’s individual characteristics, including age, gender, education, marital status, and employment status; ***fc**_i_* is the vector of agricultural migrant *i*’s migration characteristics, including migration range; and ***hc**_i_* is the vector of agricultural migrant *i*’s health characteristics, including self-reported health status; α is the intercept; *β*, *γ*, *δ*, and *η* are the regression coefficients, and *ε_i_* is the random perturbation term.

This study aimed to explain the heterogeneous effects of health insurance schemes on income and income distribution. OLS regression only examines the average effect and this method has some limitations. First, one of the preconditions of OLS regression is that the random perturbation error obeys normal distribution. However, it is difficult to fully meet the hypothesis because of the possibility of self-selection bias. Self-selection bias refer to the fact that the dependent variable is determined to some extent by individual selection [24]. The enrollment of health insurance among agricultural migrants may be determined by individual characteristics, migration characteristics, and health characteristics. And these characteristics may also have an impact on income, which leads to self-selection bias when estimating the impact of health insurance scheme on income. For the regression equation, self-selection bias means that the explanatory variable is related to the random perturbation term, which leads to the endogeneity [24]. If endogeneity exists, unadjusted OLS regression results will be biased and lead to facile inferences [25]. Second, OLS regression is valid only for the cases in which the effect of independent variables along the conditional distribution is unimportant. In that respect, as the OLS technique only reveals the impact of the different variables at the mean point of the distribution, it will be insufficient for the income distribution [26]. 

In order to solve the above problems, we adopt the propensity score matching (PSM) and quantile regression method, respectively.

PSM is one established way to reduce self-selection bias [27,28]. The logic of this approach is to match the characteristics of a treatment group (insured) with a control group (uninsured) so that their characteristics are observationally equivalent except for one crucial difference: One group decide to participate in health insurance scheme and the other group decide not to. Subsequently, if an insured gets more income, this can be attributed to the treatment effect rather than his/her characteristics (selection effects).

The treatment groups were the individuals participating in NCMS, URBMI, or UEBMI, while the control group included the individuals who did not participate in any kind of health insurance schemes. The propensity scores were estimated using three logistic regression models that examined an individual’s insurance enrollment behavior. In PSM, the dependent variable used three dummy variables indicating whether an individual had enrolled in NCMS, URBMI, or UEBMI, while the independent variables were a series of individual-level characteristics including age, gender, marital status, education, employment status, migration range, and self-reported health status. Then the appropriate matching method should be selected to match the individual in control group with the individual in the treatment group with similar propensity score. To ensure robustness of the results, we used several matching strategies, including nearest-1-neighbor matching, nearest-2-neighbor matching, and nearest-4-neighbor matching. The average treatment effect on the treated (ATT) of the health insurance on the income can be obtained by comparing the differences in outcome variables between the treatment group and the control group.

We further used the quantile regression method to capture the heterogeneous effects of health insurance schemes on income distribution. Quantile regression was preferable to other techniques in this study for two reasons. First, quantile regression mostly prefers for the income distribution, as it allows to make an estimation for specific quantiles of conditional income distribution and to describe the distribution characteristics more comprehensively. Second, quantile regression yields more robust coefficient estimates than the OLS estimates when data have both outliers and heavy-tailed distributions [29,30]. Comparisons of the regression coefficients across different percentiles allow us to infer the effects of a certain health insurance scheme at different points in the income distribution. The quantile regression model is expressed as follows:(2)$$Q_{\theta }\left(LnY_{i}|insur_{i},\;ic_{i},\;fc_{i},hc_{i}\right)=\alpha_{\theta }+\beta_{\theta }insur_{i}+\gamma_{\theta }ic_{i}+\delta_{\theta }fc_{i}+\eta_{\theta }hc_{i}+\epsilon_{i}$$
where *Q_θ_* (*Ln**Y**_i_* | ***insur**_i_*, ***ic**_i_*, ***fc**_i_*, ***hc**_i_*) denotes the *θ*th conditional quantile of the distribution of the logarithm of monthly net income with *θ* ∈ (0, 1). According to the previous studies, we estimated the quantile regressions at the 0.10, 0.25, 0.50, 0.75, and 0.90 quantile [31,32]. α*_θ_* is the intercept, *β_θ_*, *γ_θ_*, *δ_θ_*, and *η_θ_* are the regression coefficients estimated at the *θ*th quantile, and *ε_i_* is the random perturbation term. 

## 3. Results

### 3.1. Sample Characteristics

The mean of monthly net income of the 86,660 agricultural migrants was about RMB 4111. The income of the agricultural migrants in different income groups differed greatly with the gap between the highest income group (the top 10%) and the lowest income group (the bottom 10%) being RMB 9249. Out of the total 86,660 agricultural migrants, the total enrollment rate of the three health insurances was 94.07%. More than 75% of the migrants participated in NCMS, though all migrants work and live in cities. Among the whole sample, 13.88% participated in UEBMI and 2.74% participated in URBMI. The mean age of the migrants surveyed were 36 years. Out of the entire sample, more than half were male, nearly 70% received an education of middle school or below, and nearly half were the individuals migrating within the province. More than half of the respondents were employees and nearly 40% were self-employed. In addition, the vast majority of the migrants were married and with good self-reported health. 

### 3.2. Outpatient Service Utilization 

Out of the total population surveyed, 48,766 reported diseases in the last year and 18,017 sought outpatient care. Table 2 shows the proportion of the agricultural migrants seeking outpatient care when they suffered from illnesses. Chi square test was carried out for outpatient service utilization and health insurance enrollment in the whole sample and six subgroups. The results showed that most samples passed the significance test except tier 6. The outpatient service utilization of the agricultural migrants participating in UEBMI was significantly higher than that of those participating in URBMI and NCMS, especially for the middle- and low-income groups. Taking the lowest income group as an example, the outpatient service utilization of the migrants covered by UEBMI was 16.09% and 18.02% higher than that of those covered by URBMI and NCMS, respectively.

### 3.3. Effect of Health Insurance on Income and Income Distribution

We used the density plot to compare the distribution of monthly net income of agricultural migrants participating in different health insurance schemes. Figure 1 graphically represents the Kernel density estimations of monthly net income of agricultural migrants participating in three different health insurance schemes. We found that the income distributions were skewed to the right, suggesting that it was not enough to focus only on the mean effect of health insurance.

Table 3 presents the OLS regression and quantile regression results. Figure 2 summarizes the predicting effects of participating in NCMS, URBMI, or UEBMI on monthly net income at each 0.2 quantile. In order to evaluate the income effects of the three health insurance schemes, we used PSM approach at the same time. The distributions between the treatment and control groups were found to be well balanced. Table 4 reports the PSM estimation results. Although OLS model overestimated the income effects of health insurance schemes, after controlling sample selection bias and endogenous problems, URBMI and UEBMI still had significant income-increasing effects. However, participating in NCMS had no significant impact on monthly net income.

The results of OLS regression and PSM approach both showed that participating in URBMI or UEBMI increased monthly net income significantly. UEBMI played a more important role in increasing income of agricultural migrants. The quantile estimates demonstrated that participating in UEBMI significantly increased the logarithm of monthly net income by 16.73%, 13.76%, 10.52%, 9.66%, and 12.52% at the 0.10, 0.25, 0.50, 0.75, and 0.90 quantile, respectively. The income-increasing effect of UEBMI was most obvious in the low-income group. URBMI also had a significant role in increasing income of different groups, but its effect was weaker than that of UEBMI. Figure 2 indicates that the distribution effect curve of URBMI was U-shaped, i.e., URBMI had the lowest income-increasing effect on the middle-income group, but more obvious income-increasing effect on the low- and high-income group.

OLS regression found that participating in NCMS reduced monthly net income significantly. However, the effect was not significant after PSM approach was conducted. And it can be seen from quantile regression results and distribution effect curve that with the increase of income level, the income-increasing effect of NCMS decreased. But the effects were all insignificant. Therefore, enrollment in NCMS did not have income-increasing effect and did not affect income distribution.

Additionally, OLS regression showed that all the control variables had significant associations with income. Male agricultural migrants’ monthly net income was higher than that of female. Age had a significant negative correlation with income. In China, most of agricultural migrants are engaged in low-level jobs, so the correlation between age and labor output is negative. Education was found to be positively associated with income. This is because education has long be seen as an important human capital. As for marriage status, the income level of the married agricultural migrants was higher than that of the unmarried because the married are more likely to have relatively stable income sources. In terms of employment status, OLS results showed that the income level of employers and self-employed workers were both higher than that of employees. With the increase of income group, such gap became larger and larger. For the high-income group with the 90% quantile, the logarithm of monthly net income among employers was 93.57% higher than that among employees and the logarithm of monthly net income among self-employed worker was 31.75% higher than that among employees. Monthly net income of the inter-provincial migrants was significantly higher than that of the intra-provincial migrants. Inter-provincial migration usually incurs high movement cost. Most of the inter-provincial agricultural migrants are highly selective group with strong professional skills and great work ambition [33]. Self-reported health status had a significant positive association with income of the migrants, but such association decreased with the increase of income. In the low-income group corresponding to the 10% quantile, the logarithm of monthly net income among the self-reported healthy was 50.74% higher than that among the unhealthy. While in the high-income group corresponding to the 90% quantile, the logarithm of monthly net income among the self-reported healthy was only 15.80% higher than that among the unhealthy.

## 4. Discussion

This study makes an important contribution to the available literature as it is one of the very few studies on heterogeneous effects of health insurance schemes on income among agricultural migrants. We found that NCMS was the major health insurance that covered agricultural migrants in China, which is in line with a previous study indicating that NCMS is the official public health insurance for agricultural migrants in this country [34]. Our OLS and PSM results found that UEBMI and URBMI significantly increased monthly net income of agricultural migrants, while NCMS did not. And UEBMI had the highest income-increasing effect. This can be explained by the fact that UEBMI has very strict eligibility, the broadest benefit package, and the highest level of reimbursement among the three health insurance schemes. 

Meanwhile, our quantile analysis results found that UEBMI had relatively stable effects on income distribution and the income-increasing effect was larger for the poor than that for the rich. These results are in line with our other results on service use showing that the outpatient utilization of UEBMI was highest among the three public health insurance schemes. Moreover, the middle- and low-income groups covered by UEBMI used more outpatient services than those in high-income group covered by UEBMI. It is quite encouraging that UEBMI was found to have the pro-poor income distribution effect. The higher financial risk protection level of UEBMI relieves the burden of medical spending of migrants, enables them to make better use of health services, and then raises income through the increase of health and investment in other forms of capital. 

Quantile analysis also revealed that URBMI had a significant role in promoting income among subgroups of agricultural migrants and its income-increasing effect was particularly obvious for the lowest and highest income groups. However, in our study, the low- and middle-income groups were found to be with much lower outpatient service utilization than the high-income group. Especially, the lowest income group was with the lowest service use rate. The financial risk protection level of URBMI is limited with an average reimbursement rate of about 50% and it mainly covers inpatient and outpatient services for catastrophic diseases but not general outpatient services [35]. It can be inferred that though the lowest income group of population use less outpatient care, the enrollment of an urban health insurance, URBMI, strengthens their confidence to fight against disease, so they put more money into other investments, such as training, and thus increase their income level. Still, further analysis is needed to thoroughly explain the phenomenon. We also found that the income-increasing effect of URBMI was less than that of UEBMI in all income groups. This can be explained by the different benefit packages and reimbursement rates between these two insurances. 

In addition, we found that NCMS did not significantly affect income and income distribution among agricultural migrants, which is in line with previous studies indicating that NCMS did not significantly reduce health payment-induced poverty [13,14]. Our findings are partly contradictive with Qi’s previous work, which showed that NCMS significantly increased income, but did not improve income inequity within a province [18]. Compared with the other two schemes, NCMS is more designed to provide financial protection for hospitalization costs and catastrophic disease. The coverage of outpatient care of NCMS is very limited with very low reimbursement. In addition, NCMS has many restrictions on designated hospitals and rather complex reimbursement procedure. It is very inconvenient for agricultural migrants, who already work and live in urban areas, to get NCMS reimbursement [36,37]. Such reimbursement inconvenience may reduce the actual service use rate among agricultural migrants covered by NCMS. Previous studies already found that the requirement of personal payment in advance and cross-provincial reimbursement were related with the decreased health service utilization of agricultural migrants [38,39]. We also found that more than 60% of agricultural migrants reporting illness and covered by NCMS did not seek any outpatient care. With low educational level and limited professional skills, many migrants work overtime in a relatively dangerous or poor environment. Compared with urban residents, they face more health risks. Considering the very high proportion, more than 75%, of migrants covered by NCMS, the results of the low service use rate and the insignificant effect of NCMS on income and income distribution are really worrisome. 

In view of the heterogeneous effects of health insurance schemes on income and income distribution among agricultural migrants, we suggest that the public health insurance system in China needs to be reformed in two aspects. First, considering the insignificant function of NCMS in increasing income and improving income distribution, the low service use rate, as well as NCMS’s complex procedure of reimbursing care from allopatry, the government really needs to first improve the portability of NCMS, which fits the high mobility nature of agricultural migrants’ work. Then, considering that NCMS has the lowest benefit package among the three public health insurance schemes, as the major health insurance of agricultural migrants, NCMS really needs to enlarge its benefit package and thus can better benefit agricultural migrants’ health and life. In addition, considering the pro-poor nature of UEBMI in income distribution, we suggest that the government should provide more policy incentives to encourage enterprise to enroll more agricultural migrants into UEBMI and thus improving the income inequality of agricultural migrants.

Due to the limitation of our data, which did not contain information on medical spending and education and training expenditure, we were unable to discuss the internal mechanism of health insurance on income. Meanwhile, we only had outpatient service information and lacked inpatient service information in our data, which made us unable to comprehensively analyze the relationship between health service utilization and health insurance of agricultural migrants. Future research can continue to explore the impacts when better data are available. In addition, we were unable to discuss the effect of Basic Medical Insurance for Urban and Rural Residents (BMIURR). To avoid overlap in NCMS and URBMI coverage, the Chinese government formally started to integrate NCMS and URBMI into BMIURR in 2016. However, in the questionnaire survey period, when BMIURR was still in the start-up stage, the number of the insured was very small and its income effect was unable to be seen, so we did not include those covered by BMIURR in our research sample. It is worthwhile to study the income effect of BMIURR in the future.

## 5. Conclusions

Our study showed that the three public health insurance schemes in China heterogeneously affected income and income distribution among agricultural migrants. UEBMI with the broadest benefit package and highest reimbursement rate had the best effect in income increasing and showed most obvious pro-poor effect among migrants. While URBMI had a significant role in increasing income with its income-increasing effect being obvious for the lowest and highest income groups. However, NCMS, covering the most migrants, had not significant income-increasing effect. These suggest that our public health insurance system needs systematic reforms to further reduce income inequity for migrants, a very large and relatively disadvantaged sub-population in China, and thus increasing the welfare of overall population in China.

## Figures and Tables

**Figure 1 ijerph-17-03079-f001:**
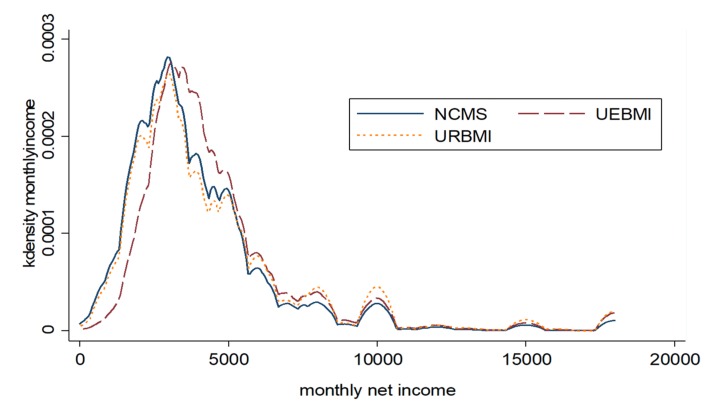
Estimated Kernel density of monthly net income distribution of agricultural migrants.

**Figure 2 ijerph-17-03079-f002:**
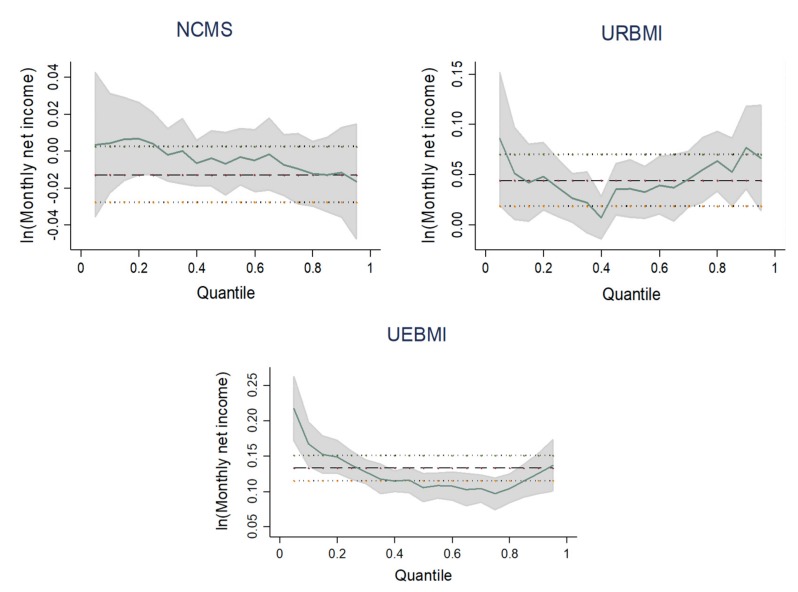
Quantile regression estimates for the three health insurance schemes.

**Table 1 ijerph-17-03079-t001:** Variables, their measurement, and descriptive statistics.

Variable	Description	Mean	SD
Monthly net income (RMB)		4111.768	2798.683
Natural logarithm of monthly net income		8.1509	0.5769
Member of NCMS	1 if yes, 0 if not	0.7745	0.4179
Member of URBMI	1 if yes, 0 if not	0.0274	0.1633
Member of UEBMI	1 if yes, 0 if not	0.1388	0.3457
Age (Year)		36.0530	9.8420
Gender	1 if male, 0 if female	0.5773	0.4940
Education	1 if primary or below; 2 if middle school; 3 if senior school; 4 if college or above	2.2425	0.8764
Marital status	1 if married, 0 if unmarried	0.8146	0.3886
Employment status	1 if employee; 2 if employer; 3 if self-employed; 4 if other	1.8564	0.9883
Migration range	1 if intra-provincial migration; 0 if inter-provincial migration	0.4864	0.4998
Self-reported health status	1 if good; 0 if poor	0.9865	0.1155

**Table 2 ijerph-17-03079-t002:** Outpatient service utilization of the agricultural migrants with different types of health insurances (%).

Proportion of Outpatient Treatment							
	Total	Tier 1	Tier 2	Tier 3	Tier 4	Tier 5	Tier 6
UEBMI	42.62	43.68	41.08	42.27	44.48	43.16	39.84
URBMI	36.30	27.59	32.97	34.68	34.17	46.98	43.88
NCMS	35.88	25.66	33.82	36.79	37.32	37.44	39.23
χ^2^ (Sig.)	126.01 (<0.001)	13.93 (<0.001)	25.46 (<0.001)	24.13 (<0.001)	45.35 (<0.001)	14.50 (<0.001)	1.71 (>0.100)

Note: Tier 1 to Tier 6 represent the bottom 10%, 10–25%, 25–50%, 50–75%, 75–90%, and top 10% of the income distribution, respectively.

**Table 3 ijerph-17-03079-t003:** The results of ordinary least squares (OLS) and quantile regressions (QR).

	OLS	Quantile				
		0.10	0.25	0.50	0.75	0.90
NCMS	−0.0127 *	0.0042	0.0041	−0.0068	−0.0095	−0.0115
	(0.0077)	(0.0149)	(0.0093)	(0.0088)	(0.0097)	(0.0140)
URBMI	0.0443 ***	0.0512 **	0.0368 **	0.0361 **	0.0549 ***	0.0767 ***
	(0.0131)	(0.0255)	(0.0160)	(0.0151)	(0.0166)	(0.0239)
UEBMI	0.1328 ***	0.1673 ***	0.1376 ***	0.1052 ***	0.0966 ***	0.1252 ***
	(0.0091)	(0.0177)	(0.0111)	(0.0104)	(0.0115)	(0.0165)
Age	−0.0064 ***	−0.0071 ***	−0.0061 ***	−0.0061 ***	−0.0050 ***	−0.0046 ***
	(0.0002)	(0.0004)	(0.0003)	(0.0003)	(0.0003)	(0.0004)
Gender	0.2780 ***	0.2769 ***	0.2890 ***	0.2921 ***	0.2872 ***	0.3172 ***
	(0.0037)	(0.0072)	(0.0045)	(0.0042)	(0.0047)	(0.0067)
Education						
Middle school	0.0999 ***	0.1776 ***	0.1042 ***	0.0753 ***	0.0637 ***	0.0862 ***
	(0.0051)	(0.0100)	(0.0063)	(0.0059)	(0.0065)	(0.0094)
Senior school	0.1670 ***	0.2305 ***	0.1496 **	0.1266 ***	0.1267 ***	0.1583 ***
	(0.0062)	(0.0122)	(0.0076)	(0.0072)	(0.0079)	(0.0114)
College or above	0.2828 ***	0.3204 ***	0.2535 ***	0.2338 ***	0.2724 ***	0.3300 ***
	(0.0078)	(0.0152)	(0.0095)	(0.0090)	(0.0099)	(0.0142)
Marital status	0.1731 ***	0.1744 ***	0.1443 ***	0.1556 ***	0.1686 ***	0.1829 ***
	(0.0052)	(0.0101)	(0.0063)	(0.0059)	(0.0066)	(0.0095)
Employment status						
Employer	0.5136 ***	0.1929 ***	0.3261 ***	0.4646 ***	0.7086 ***	0.9357 ***
	(0.0082)	(0.0160)	(0.0100)	(0.0094)	(0.0104)	(0.0150)
Self-employed worker	0.0758 ***	−0.1697 ***	−0.0463 ***	0.0736 ***	0.1886 ***	0.3175 ***
	(0.0040)	(0.0078)	(0.0049)	(0.0046)	(0.0051)	(0.0074)
Other	−0.1116 ***	−0.2948 ***	−0.1775 ***	−0.1070 ***	−0.0395 **	0.0652 **
	(0.0139)	(0.0272)	(0.0170)	(0.0160)	(0.0177)	(0.0254)
Migration range	−0.1409 ***	−0.1458 ***	−0.1394 ***	−0.1411 **	−0.1441 ***	−0.1625 ***
	(0.0036)	(0.0070)	(0.0044)	(0.0041)	(0.0046)	(0.0066)
Self-reported health status	0.2679 ***	0.5074 ***	0.3221 ***	0.2460 ***	0.2074 **	0.1580 ***
	(0.0156)	(0.0304)	(0.0190)	(0.0180)	(0.0198)	(0.0285)
Constant	7.7054 ***	6.9159 ***	7.4034 ***	7.7588 ***	8.0086 ***	8.2432 ***
	(0.0197)	(0.0384)	(0.0240)	(0.0226)	(0.0249)	(0.0359)
Adj R-squared	0.1685					

Note: Standard errors in parentheses; * *p* < 0.1, ** *p* < 0.05, *** *p* < 0.01.

**Table 4 ijerph-17-03079-t004:** Average treatment effect on the treated (ATT) of the health insurance schemes on monthly net income.

	NCMS		URBMI		UEBMI	
	ATT	*t*-Stat	ATT	*t*-Stat	ATT	*t*-Stat
nearest-1-neighbor PSM	−0.0015	−0.13	0.0349 *	1.79	0.1226 ***	6.60
nearest-2-neighbor PSM	0.0050	0.45	0.0453 **	2.34	0.1321 ***	7.14
nearest-4-neighbor PSM	−0.0010	−0.06	0.0237	1.28	0.1178 ***	6.64

Note: * *p* < 0.1, ** *p* < 0.05, *** *p* < 0.01.

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
