# Peer review of "Do Health Insurance Schemes Heterogeneously Affect Income and Income Distribution? Evidence from Chinese Agricultural Migrants Survey"

_ijerph, 2020, doi:10.3390/ijerph17093079_

Round 1

Reviewer 1 Report

Dear author(s), i found very interesting your studi and your topic! In my opinion the paper is valid. Is necessary to improve some MINOR concern:

In the introduction is necesssary to explain better whic are the research gaps, please do this thing.

Methodology section i really appreciate this section, the only raccomandation regards the explanation of propensity score.
Conclusion section is too short please improve this section.

Author Response

Reviewer 1

Dear author(s), i found very interesting your studi and your topic! In my opinion the paper is valid. Is necessary to improve some MINOR concern:

  1. In the introduction is necesssary to explain better whic are the research gaps, please do this thing.

We added an explanation of the research gaps in the introduction. In our opinion, there are mainly three research gaps:

(1)The exiting studies on income effect of health insurance focused on the average income effect, but did not further analyze the income distribution effect of health insurance for subgroups.

(2)To our knowledge, very few studies have existed on the heterogeneous effect of different health insurance schemes on the related outcomes, especially income.

(3)The study of health insurance effect on income among agricultural migrants in China is very scarce. Because agricultural migrants in China have heterogeneous accesses to the three public health insurance schemes, comparing the income effects of the three schemes among them will be very interesting and policy relevant.

  1. Methodology section i really appreciate this section, the only raccomandation regards the explanation of propensity score.

We added an explanation of propensity score matching (PSM) in the empirical analysis part. The main purpose of using PSM is to reduce the endogeneity caused by self-selection bias. If endogeneity exists, unadjusted OLS results will be biased and lead to facile inferences. The logic of PSM is to match the characteristics of a treatment group (insured) with a control group (uninsured) so that their characteristics are observationally equivalent except for one crucial difference: one group decide to participate in health insurance scheme and the other group decide not to. Subsequently, if an insured gets more income, this can be attributed to the treatment effect rather than his/her characteristics (selection effects).

  1. Conclusion section is too short please improve this section.

We added some sentences in the conclusion section.

Reviewer 2 Report

This is a well-written paper with an important message, because the manuscript has an important message, I need to make sure authors covered all types of possible biases, here are my suggestions:

  1. Please add a density plot and compare the distribution of income or ln(income) across three different HI plan, looking at the T3 NCMS coeff. in different quantiles show that NCMS data more likely be a normally distributed data, but I like to see that, also looking at the Fig1 we can see the same message that OLS is the best fit model for NCMS.
  2. Is it possible someone has two types of HI, for example NCMS and URBMI or URBMI? If not please explain that.
  3. How did you control the regressions by location, in somehow the type of insurance shows the location (because two of them are for urban areas and one for rural) but try the location on the model and see the results, I would like to see how that variable (location) changes the coeff. direction for the NCMS, by default rural areas receive lower payment or income, it may be a reason for the negative coeff.
  4. Have you weighted the models? I would like to see how number of urban and rural pop change the coeff., if you are not able to ran a weighted model, please explain that.
  5. Please add a short discussion on negative OLS for the NCMS, it is hard to believe the NCMS has negative coeff in comparison with non-HI or maybe non-insured population are rich population because this is a subsidize insurance and cover the poor communities.
  6. Formula in p4 line 137: You should add another error term for the quantiles, please see the mostly harmless econometric books for more information, however it is up to authors to keep the current formula or add another error term.

Author Response

Reviewer 2

This is a well-written paper with an important message, because the manuscript has an important message, I need to make sure authors covered all types of possible biases, here are my suggestions:

  1. Please add a density plot and compare the distribution of income or ln(income) across three different HI plan, looking at the T3 NCMS coeff. in different quantiles show that NCMS data more likely be a normally distributed data, but I like to see that, also looking at the Fig1 we can see the same message that OLS is the best fit model for NCMS.

We used the density plot to compare the distributions of monthly net income of agricultural migrants participating in different health insurances. Density plot indicated that the income distributions were skewed to the left.

We added the above analysis in 3.3. Effect of health insurance on income and income distribution.

  1. Is it possible someone has two types of HI, for example NCMS and URBMI or URBMI? If not please explain that.

Yes, it is. Agricultural migrants can not only participate in NCMS in rural areas, but also participate in urban insurances in urban areas. But the number of repeat insured is very small. In order to measure the income effect of a single type of public health insurance scheme effectively, we ruled out those who had more than one public health insurances and included 86,660 individuals in the final sample. We explained this in 2.1. Dataset.

  1. How did you control the regressions by location, in somehow the type of insurance shows the location (because two of them are for urban areas and one for rural) but try the location on the model and see the results, I would like to see how that variable (location) changes the coeff. direction for the NCMS, by default rural areas receive lower payment or income, it may be a reason for the negative coeff.

The samples we chose were all migrants that had entered cities to live. All the survey sites are urban areas. So we didn’t control the regressions by location.

In order to make the reviewers and readers to better understand the sample characteristics, we added the related information in 2.1. Dataset.

  1. Have you weighted the models? I would like to see how number of urban and rural pop change the coeff., if you are not able to ran a weighted model, please explain that.

We didn’t weight the model because that the samples we used were all migrants from rural areas. We added the basic characteristics of the sample in 2.1. Dataset.

  1. Please add a short discussion on negative OLS for the NCMS, it is hard to believe the NCMS has negative coeff in comparison with non-HI or maybe non-insured population are rich population because this is a subsidize insurance and cover the poor communities.

We added more regarding the effect of NCMS in the discussion section and in the methods section. Due to self-selection bias, we think that the results of OLS regression may be biased (the explanation of this problem is supplemented in 2.3. Empirical analysis). For this reason, we conducted PSM test, expecting to further confirm the income effect of NCMS. The results of PSM showed that NCMS had no significant effect. Our explanation for this is that in the Chinese context, the medical expenses incurred by agricultural migrants living in cities can be reimbursed through NCMS, but the reimbursement procedure is complex and the reimbursement proportion is low. It may make many agricultural migrants choose not to use NCMS, resulting in the income effect of NCMS is not obvious. Previous studies already found that the requirement of personal payment in advance and cross-provincial reimbursement were related with the decreased health service utilization of agricultural migrants.

  1. Formula in p4 line 137: You should add another error term for the quantiles, please see the mostly harmless econometric books for more information, however it is up to authors to keep the current formula or add another error term.

We have read the literature of quantile regression and modified the formula (see formula 2).

Reviewer 3 Report

why you have used three methods of analyses; ols, quantile regression and PSM? any particular reason!

What if you, instead, categorize the income into low middle and upper income bracket and try to run either simple OLS or PSM and see if these insurance schemes have differential impact on various income groups.

Why you dont try to run a separate regression based on gender?

I suspect migration to places farther away from home that is other states will have serious implication for health insurance scheme adoption and its impact on income hence it is suggested if you can look into the distance to home as one of the variable in your regression or try to run a separate regression on sample of migrants who migrated within the state and those who migrated outside the state.

Author Response

Reviewer 3

  1. why you have used three methods of analyses; ols, quantile regression and PSM? any particular reason!

We have added the related explanations in 2.3. Empirical analysis. The purpose of OLS regression is to test the average income effect of three different health insurance schemes. Due to self-selection bias, we think that the results of OLS regression may be biased. For this reason, we conducted PSM test, expecting to further confirm the income effect of NCMS. Quantile regression can overcome the limitations of the OLS regression and evaluate heterogeneous effects of health insurance schemes on different income groups.

  1. What if you, instead, categorize the income into low middle and upper income bracket and try to run either simple OLS or PSM and see if these insurance schemes have differential impact on various income groups.

We believe that quantile regression has more advantages than regression after dividing income into low middle and upper income bracket.

We have added the related explanations in 2.3. Empirical analysis. The quantile regression can obtain a much more complete view of the effects of explanatory variables on the dependent variable and it is more robust when the random perturbation term is not normal distribution and there are outliers. Comparisons of the regression coefficients across different percentiles allowed us to infer the effects of a certain health insurance scheme at different points in the income distribution.

  1. Why you dont try to run a separate regression based on gender?

We found that gender had a significant impact on income (Table 3). However, this study focused on the income effect of health insurance. We took gender as a control variable and brought it into the regression equation. The purpose was to analyze the income effect of the three health insurance schemes when controlling gender. So we didn’t run a separate regression based on gender.

  1. I suspect migration to places farther away from home that is other states will have serious implication for health insurance scheme adoption and its impact on income hence it is suggested if you can look into the distance to home as one of the variable in your regression or try to run a separate regression on sample of migrants who migrated within the state and those who migrated outside the state.

In the survey, we could not get the information on the distance to home, so we used whether to migrate within/outside of the originally resided province (i.e. migration range) to represent the distance to home. We found that migration range had a significant impact on income (Table 3) and explained it in 3. Results. However, this study focused on the income effect of health insurance. We took migration range as a control variable and brought it into the regression equation. The purpose was to analyze the income effect of three health insurance schemes when controlling migration range. So we didn’t run a separate regression based on migration range.

Reviewer 4 Report

In my opinion, the contribution of this paper is rather limited
and I am not fully convinced if the grade of novelty is high enough to merit publication in
IJERPH. There are some major and minor remarks:
1. While the literature part of the paper is well written and quite interesting, the method-
ology part is insufficient. Starting with OLS, where the explanations of the regressors
are missing. It is not clear if it is a univariate or a multiple regression model, and there
is no value range for i. The authors should mark vectors and matrices in boldface.
What are the dimensions of the vectors and matrices? What are the assumptions
regarding the noise term? Is it normally distributed or "only" i.i.d.? Why did the
authors choose a log-level model? The paper is quiet about these questions.
2. The same applies to the quantile regression. I see no reason why we should take OLS
and then quantile regression. What is the motivation for using a quantile regression?
As above, the model has to be explained and motivated in more detail. How did the
authors choose the specific quantiles? By the way, Alpha is also a regression coefficient.
3. Unfortunately, there are some mistakes: grammatical, spelling, stylistic etc. There are
some unreadable sentences in the paper, e.g. in line 22 and 115. Sometimes there is
space between a word and a bracket, sometimes not. The authors should check and
revise English language and style carefully.
4. The abstract is partially written in past form which seems odd.

Author Response

Reviewer 4

In my opinion, the contribution of this paper is rather limited
and I am not fully convinced if the grade of novelty is high enough to merit publication in
IJERPH. There are some major and minor remarks:

  1. While the literature part of the paper is well written and quite interesting, the method-
    ology part is insufficient. Starting with OLS, where the explanations of the regressors
    are missing. It is not clear if it is a univariate or a multiple regression model, and there
    is no value range for i. The authors should mark vectors and matrices in boldface.
    What are the dimensions of the vectors and mratrices? What are the assumptions
    regarding the noise term? Is it normally distributed or "only" i.i.d.? Why did the
    authors choose a log-level model? The paper is quiet about these questions.

We added extended explanations of the method in the paper. Some specific issues are listed as follows:

We have made it clear that our OLS regression is a multiple regression model in the paper. The key independent variables were expressed in three dummy variables to indicate whether agricultural migrants participated in NCMS, URBMI, or UEBMI. The other explanatory variables, i.e. control variables, were classified into three categories: individual characteristics, migration characteristics, and health characteristics.

i in OLS formula represents each migrant. 

Regarding the assumption of noise term, we used the density plot to represent the distribution of dependent variable. And the results showed that the income distributions were skewed to the left (We added the above analysis in 3.3. Effect of health insurance on income and income distribution). Considering that the random perturbation error in OLS was impossible to obey normal distribution in our study, on one hand, we transformed the dependent variable into natural logarithm and made it close to a normal distribution, on the other hand, we tested the robustness and solved self-selection bias through propensity score matching (PSM). We have added the related explanations in 2.3. Empirical analysis.

Why did we choose a log-level model? We referred to Keynes’s research and hoped to make the dependent variable close to normal distribution through logarithmic transformation. We have added explanations in 2.3. Empirical analysis.

  1. The same applies to the quantile regression. I see no reason why we should take OLS
    and then quantile regression. What is the motivation for using a quantile regression?
    As above, the model has to be explained and motivated in more detail. How did the
    authors choose the specific quantiles? By the way, Alpha is also a regression coefficient.

The density plot indicated that the income distributions were skewed to the left. So we used quantile regression to overcome the limitations of the OLS regression.

The quantile regression can obtain a much more complete view of the effects of explanatory variables on the dependent variable and it is more robust when the random perturbation term is not normal distribution and there are outliers. Comparisons of the regression coefficients across different percentiles allowed us to infer the effects of a certain health insurance scheme at different points in the income distribution. We have added the explanation in 2.3. Empirical analysis.

According to the previous studies, we estimated the quantile regressions at the 0.10, 0.25, 0.50, 0.75, and 0.90 quantile. We added the reference literature in 2.3. Empirical analysis.

In addition, we have modified the formula (1) and (2).

  1. Unfortunately, there are some mistakes: grammatical, spelling, stylistic etc. There are
    some unreadable sentences in the paper, e.g. in line 22 and 115. Sometimes there is
    space between a word and a bracket, sometimes not. The authors should check and
    revise English language and style carefully.

We have changed the sentence in line 22 and we deleted the sentence in line 115. We changed a lot in styles of space, bracket, tables and so on. We also corrected some English typos. At the same time, we changed some sentences to make them clear to the author.

  1. The abstract is partially written in past form which seems odd.

We have rephrased a few sentences in the abstract. The first sentence in the abstract was in present tense because it is the present situation of migrant workers in China. The last sentence in the abstract was also in present tense because that was our suggestions. We used present tense in similar situations of our previous published papers. However, we believe that there are other ways of dealing with tense in abstract.

Round 2

Reviewer 3 Report

authors have incorporated most of the concerns of reviewers hence paper can be accepted for publication.

Author Response

We have checked English language and style of our papers and made minor revisions. 

Reviewer 4 Report

Unfortunately, I am not fully satisfied with the revisions. They seem to have been made in a
hurry. Some of my suggestions in the reviewer report have not been incorporated to improve
the paper accurately. Moreover, the authors did not provide a revised version of the paper
where changes are highlighted, which is disadvantageous. There are still some additions and
motivations needed, the methodology part is still not fully sufficient. Moreover, I am still
of the opinion that the contribution of this paper is rather limited and I am not convinced
if the grade of novelty is high enough to merit publication in IJERPH. The authors should
read my suggestions of the previous report carefully. Moreover, there are still some mistakes
and the "corrected" sentences are still unreadable. As for the abstract, the authors have
misunderstood me, the past tense is unsuitable in my opinion.

Author Response

Reviewer 4’s comments in the second round were as follows:

Unfortunately, I am not fully satisfied with the revisions. They seem to have been made in a hurry. Some of my suggestions in the reviewer report have not been incorporated to improve the paper accurately. Moreover, the authors did not provide a revised version of the paper where changes are highlighted, which is disadvantageous. There are still some additions and motivations needed, the methodology part is still not fully sufficient. Moreover, I am still of the opinion that the contribution of this paper is rather limited and I am not convinced if the grade of novelty is high enough to merit publication in IJERPH. The authors should read my suggestions of the previous report carefully. Moreover, there are still some mistakes and the "corrected" sentences are still unreadable. As for the abstract, the authors have misunderstood me, the past tense is unsuitable in my opinion.

Considering the comments above, we have readdressed his/her comments in the first round:

1.While the literature part of the paper is well written and quite interesting, the methodology part is insufficient. Starting with OLS, where the explanations of the regressors are missing. It is not clear if it is a univariate or a multiple regression model, and there is no value range for i. The authors should mark vectors and matrices in boldface. What are the dimensions of the vectors and matrices? What are the assumptions regarding the noise term? Is it normally distributed or "only" i.i.d.? Why did the authors choose a log-level model? The paper is quiet about these questions.

We have made it clear that our OLS regression is a multiple regression model in the first revised version. The key independent variables were expressed in three dummy variables to indicate whether agricultural migrants participated in NCMS, URBMI, or UEBMI. The other explanatory variables, i.e. control variables, were classified into three categories: individual characteristics, migration characteristics, and health characteristics.

i in OLS formula represents each migrant, and we marked the value range of i in the latest revised version (i=1, 2,……86660).

We also marked vectors in the formula (1) in boldface in the latest revised version and explained the specific indicators of the vectors. However, we seldom find that the matrix form is used to express OLS regression equation in the same literatures, so we did not add matrix related content in the latest revised version.

Regarding the assumption of noise term, we used the density plot to represent the distribution of dependent variable. And the results showed that the income distributions were skewed to the right We added the above analysis in 3.3. Effect of health insurance on income and income distribution in the first revised version.

Why did we choose a log-level model? We referred to Keynes’s research and hoped to make the dependent variable close to normal distribution through logarithmic transformation. We added explanations in 2.3. Empirical analysis in the first revised version.

  1. The same applies to the quantile regression. I see no reason why we should take OLS and then quantile regression. What is the motivation for using a quantile regression? As above, the model has to be explained and motivated in more detail. How did the authors choose the specific quantiles? By the way, Alpha is also a regression coefficient.

We added OLS regression’s limitations in the latest revised version.

First, one of the preconditions of OLS regression is that the random perturbation error obeys normal distribution. However, our data did not meet this hypothesis because of self-selection bias. We solved self-selection bias through propensity score matching (PSM).

Second, OLS regression is valid only for the cases in which the effect of independent variables along the conditional distribution is unimportant. In that respect, as the OLS technique only reveals the impact of the different variables at the mean point of the distribution, it will be insufficient for the income distributions. So we used quantile regression. Quantile regression was preferable to other techniques in this study for two reasons. On one hand, quantile regression fits well for the analysis of income distribution, as it allows to make an estimation for specific quantiles of conditional income distribution and to describe the distribution characteristics more comprehensively. One the other hand, quantile regression yields more robust coefficient estimates than OLS estimates when data have outliers and heavy-tailed distributions.

According to the previous studies, we estimated the quantile regressions at the 0.10, 0.25, 0.50, 0.75, and 0.90 quantile. We added the reference literature in 2.3. Empirical analysis in the first revised version.

We marked the αθ is the intercept in the first revised version and also marked vectors in the formula (2) in boldface in the latest revised version.

  1. There are still some additions and motivations needed, the methodology part is still not fully sufficient.

Based on the answers to the above questions, we have added a supplementary description of the methodology in the latest revised version.

We explained the two limitations of OLS regression. The main purpose of using PSM is to reduce the endogeneity caused by self-selection bias and to confirm the income effect of health insurance schemes. If endogeneity exists, unadjusted OLS results will be biased and lead to facile inferences. Quantile regression can obtain a much more complete view of the heterogeneous effects of health insurance schemes on different income groups and it is more robust when the random perturbation term is not normal distribution and there are outliers.

In addition, in the second round, Reviewer 4 commented that “Moreover, there are still some mistakes and the "corrected" sentences are still unreadable. As for the abstract, the authors have misunderstood me, the past tense is unsuitable in my opinion”.

One of the sentences Reviewer 4 thought it unreadable was deleted in the resubmission last time. We rephrased the other “unreadable” sentence this time. In addition, we changed the tense in the abstract.

Round 3

Reviewer 4 Report

Thank you for considering my remarks.